# Screening of Gas Substrate and Medium Effects on 2,3-Butanediol Production with *C. ljungdahlii* and *C. autoethanogenum* Aided by Improved Autotrophic Cultivation Technique

**Luca Ricci [1,2,\*], Valeria Agostino [1], Debora Fino [1,2] and Angela Re [1,\*]**

[1] Centre for Sustainable Future Technologies, Fondazione Istituto Italiano di Tecnologia, Via Livorno 60, 10144 Turin, Italy; valeria.agostino@iit.it (V.A.); debora.fino@polito.it (D.F.)

[2] Department of Applied Science and Technology, Politecnico di Torino, Corso Duca degli Abruzzi 24, 10129 Turin, Italy

\* Correspondence: luca.ricci@iit.it (L.R.); angela.re@iit.it (A.R.)

**Abstract:** Gas fermentation by acetogens of the genus *Clostridium* is an attractive technology since it affords the production of biochemicals and biofuels from industrial waste gases while contributing to mitigate the carbon cycle alterations. The acetogenic model organisms *C. ljungdahlii* and *C. autoethanogenum* have already been used in large scale industrial fermentations. Among the natural products, ethanol production has already attained industrial scale. However, some acetogens are also natural producers of 2,3-butanediol (2,3-BDO), a platform chemical of relevant industrial interest. Here, we have developed a lab-scale screening campaign with the aim of enhancing 2,3-BDO production. Our study generated comparable data on growth and 2,3-BDO production of several batch gas fermentations using *C. ljungdahlii* and *C. autoethanogenum* grown on different gas substrates of primary applicative interest ($CO_2 \cdot H_2$, $CO \cdot CO_2$, syngas) and on different media featuring different compositions as regards trace metals, mineral elements and vitamins. $CO \cdot CO_2$ fermentation was found to be preferable for the production of 2,3-BDO, and a fair comparison of the strains cultivated in comparable conditions revealed that *C. ljungdahlii* produced 3.43-fold higher titer of 2,3-BDO compared to *C. autoethanogenum*. Screening of different medium compositions revealed that mineral elements, Zinc and Iron exert a major positive influence on 2,3-BDO titer and productivity. Moreover, the $CO_2$ influence on CO fermentation was explored by characterizing *C. ljungdahlii* response with respect to different gas ratios in the $CO \cdot CO_2$ gas mixtures. The screening strategies undertaken in this study led to the production of $2.03 \pm 0.05$ g/L of 2,3-BDO, which is unprecedented in serum bottle experiments.

**Keywords:** gas fermentation; acetogen; 2,3-butanediol; gaseous substrate; medium optimization; *C. ljungdahlii*; *C. autoethanogenum*

## 1. Introduction

Gas fermentation mediated by acetogenic bacteria of the genus *Clostridium* enables carbon capture and conversion of greenhouse gases into useful products [1–4]. Therefore, the gas fermentation technology meets a two-fold objective, contributing to counteract the effects of the current carbon cycle alterations and, simultaneously, embodying carbon circular industrial biomanufacturing models [5]. Gas fermentation technology offers a unique level of feedstock flexibility. In fact, the gaseous feedstock may be the direct waste streams of industrial processes, such as steel manufacturing and oil refineries, or syngas of variable composition generated from gasification of an ample range of carbon-containing wastes [6]. To reflect such a high feedstocks' flexibility, the synthetic gas mixtures used to grow acetogens in lab-scale gas fermentations are highly diversified and include $CO \cdot CO_2$ [7–9],

$CO_2 \cdot H_2$ [10–12] and syngas [6,13,14]. Compared to conventional chemocatalytic or thermochemical processes, gas fermentation benefits include milder process conditions, higher product selectivity owing to high enzymatic conversion specificities, higher tolerance to impurities which are known to poison inorganic catalysts [2], and higher flexibility to the employed feedstocks [15,16]. Nevertheless, the gas fermentation primary limitations are the poor gas to liquid mass transfer, due to the low solubility of the gas substrates [17], by the slow growth of acetogens that strive at the thermodynamic edge of life [18] and by low product yields [19].

Nonetheless, these challenging aspects do not represent unsolvable barriers to the use of gas fermentation as a commercial production system. In fact, industrial-scale processes have already been developed for ethanol production [20–22]. Although industrially relevant process parameters have not been achieved yet, process scalability has also been proven for the production of 2,3-butanediol (2,3-BDO), with a productivity of up to 45 g/L/d achieved by Simpson and co-workers [23]. 2,3-BDO is a valuable platform chemical [24], which can be used to produce butadiene, acetoin, diacetyl, methyl ethyl ketone and as a novel chain initiator and extender of polyurethane foams [25]. Acetogens of the genus *Clostridium* are common biocatalysts for gas fermentation applications since they feature among the highest growth rates and are natural producers of metabolites of commercial interest, such as ethanol and 2,3-BDO [26,27]. *Clostridium ljungdahlii* [28,29] and *Clostridium autoethanogenum* [30,31] are the most intensively investigated acetogenic biocatalysts.

The identification of the biocatalyst(s) and gaseous substrate(s) suitable for the production of the target product is paramount to ensure the viability of the entire process. Some studies were reliant on a single acetogen and explored its growth and product profile against several gaseous substrates. This is the case for *C. autoethanogenum* [12,32–35] and for *C. ljungdahlii* [7–9,36]. In some other studies, several strains were comparatively screened against a certain gaseous substrate [26,27,37–39]. Moreover, the identification of process conditions that act to enhance culture growth rates and/or productivities is necessary to improve the biological conversion profitability. Therefore, several studies focused on the screening of different media using a single strain [40,41] or multiple strains [37]. In spite of the undoubtful value of each study, gaining a thorough understanding of the sensitivity of *C. autoethanogenum* and *C. ljungdahlii* growth and product profile to variations in the gaseous substrate and/or medium is puzzling. This is severely hindered by the fact that the data on autotrophic growth and product formation are rarely acquired in comparable cultivation conditions [38].

Our study acquired comparable data on the growth and metabolite profile of a wide range of batch gas fermentations carried out by *C. autoethanogenum* and *C. ljungdahlii* in order to individuate the gas substrates and medium compositions that afford enhanced 2,3-BDO production. To this aim, both the model acetogens were grown on several gaseous substrates, which explored the inclusion of different gases or the usage of different relative gas ratios, and on four media featuring different compositions as regards trace metals, mineral elements and vitamins. Further provided is a pre-screening aimed at individuating a cultivation technique able to enhance 2,3-BDO titer and productivity. More precisely, the screening campaign was structured as follows. Firstly, we explored the cultivation technique for autotrophic batch experiments in serum bottles, by comparing the results obtained using two different volumetric gas to liquid ratios (4:1 and 9:1) and two different bottle orientations during agitation (vertical and horizontal orientation). We then identified the gas substrate and biocatalyst which promoted the production of 2,3-BDO by screening growth and product profiles of *C. autoethanogenum* and *C. ljungdahlii* on three gas substrates—$CO_2 \cdot H_2$, $CO \cdot CO_2$ and simulated syngas. Subsequently, we investigated the influence of medium composition on 2,3-BDO titers and productivities by testing different media with the previously selected biocatalyst and gas substrate. Finally, the study of the 2,3-BDO production process encompassed a further level of investigation of the gas substrate, by exploring how different gas ratios impact on the metabolites' profile.

## 2. Materials and Methods

### 2.1. Bacterial Strains and Medium Compositions

*C. autoethanogenum* DSM 10061 and *C. ljungdahlii* DSM 13582 were purchased from DSMZ (Deutsche Sammlung von Mikroorganismen und Zellkulturen GmbH, Braunschweig, Germany) and cultivated under strictly anaerobic conditions. To characterize growth and production profiles of the two acetogenic bacteria, four different cultivation media were employed. The "Tanner_mod" medium is a modified Tanner medium [42] as it is based on the medium proposed by Bengelsdorf and co-workers [27] and foresees modified concentrations of 2-(N-morpholino)ethanesulfonic acid (MES), $ZnSO_4$, $Na_2SeO_3$, vitamin B12 and omits sodium-2-mercaptoethansulfonate (MESNA) (Table 1). The "Valgepea_mod" medium is a modified Valgepea medium as it is based on the medium introduced by Valgepea and co-workers [13] but foresees the addition of MES and different cysteine and resazurin concentrations (Table 1). The "Tan_Val" medium is introduced for the first time in this study and results from the combination of the mineral solution of the "Tanner_mod" medium with the trace element solution and vitamin solution of the "Valgepea_mod" medium. Finally, the "Tan_Val + Fe" medium differs from the "Tan_Val" medium only by the addition of 0.017 g/L of $FeCl_3 \times 6 H_2O$. The detailed recipes and compositions of the four media used in this study are reported in Table 1. The pH of each medium employed was adjusted to 5.9 with NaOH 10 M. Under heterotrophic growth conditions, 5 g/L of fructose was added in each medium. All chemicals employed in this study were purchased from Sigma-Aldrich (St. Louis, MO, USA), Alpha Aesar (Haverhill, MA, United States), Fisher BioReagents by Thermo Fisher Scientific (Waltham, MA, USA), Carl Roth GmbH + Co. KG (Karlsruhe, Germany), Janssen Pharmaceuticals (Geel, Belgium), and Merck KGaA (Damstadt, Germany).

**Table 1.** Overview of media compositions. In the table are reported the compositions of all the media tested in this study, in gram (g), milligram (mg) of milliliter (mL) per liter of solutions. * These elements have been added through the mineral solution. ** These elements, in the "Valgepea_mod" medium, have been added directly to the medium.

| Medium Composition (Per Liter) | | | | | |
|---|---|---|---|---|---|
| | Tanner_Mod | Valgepea_Mod | Tan_Val | Tan_Val + Fe | |
| MES | 10 | 10 | 10 | 10 | g |
| yeast extract | 0.5 | / | / | / | g |
| resazurin (1 g/L) | 1 | 1 | 1 | 1 | mL |
| cysteine-HCl*$H_2O$ | 1 | 1 | 1 | 1 | g |
| Mineral solution | 25 | / | 25 | 25 | mL |
| Trace element solution | 10 | 10 | 10 | 10 | mL |
| Vitamin solution | 10 | 10 | 10 | 10 | mL |
| NaCl | * | 0.2 | * | * | g |
| $NH_4Cl$ | * | 2.5 | * | * | g |
| KCl | * | 0.5 | * | * | g |
| $NaH_2PO_4$*$H_2O$ | * | 2.3439 | * | * | g |
| $MgCl_2$*$6H_2O$ | * | 0.5 | * | * | g |
| $CaCl_2$*$2H_2O$ | * | 0.1324 | * | * | g |
| $FeCl_3$*$6H_2O$ | * | 0.017 | * | 0.017 | g |
| Mineral Solution (Per Liter) | | | | | |
| | Tanner_Mod | Valgepea_Mod | Tan_Val | Tan_Val + Fe | |
| NaCl | 80 | ** | 80 | 80 | g |
| $NH_4Cl$ | 100 | ** | 100 | 100 | g |
| KCl | 10 | ** | 10 | 10 | g |
| $KH_2PO_4$ | 10 | ** | 10 | 10 | g |
| $MgSO_4$*$7H_2O$ | 20 | ** | 20 | 20 | g |
| $CaCl_2$*$2H_2O$ | 4 | ** | 4 | 4 | g |

**Table 1.** *Cont.*

| Trace Element Solution (Per Liter) | | | | | |
| --- | --- | --- | --- | --- | --- |
| | **Tanner_Mod** | **Valgepea_Mod** | **Tan_Val** | **Tan_Val + Fe** | |
| Nitrilotriacetic acid | 2 | 1.5 | 1.5 | 1.5 | g |
| $MnSO_4*H_2O$ | 1 | 0.5 | 0.5 | 0.5 | g |
| $Fe(SO_4)_2(NH_4)_2*6H_2O$ | 0.8 | / | / | / | g |
| $FeSO_4*7H_2O$ | / | 0.667 | 0.667 | 0.667 | g |
| $CoCl_2*6H_2O$ | 0.2 | 0.2 | 0.2 | 0.2 | g |
| $ZnSO_4*7H_2O$ | 0.001 | 0.2 | 0.2 | 0.2 | g |
| $CuCl_2*2H_2O$ | 0.02 | 0.02 | 0.02 | 0.02 | g |
| $NiCl_2*6H_2O$ | 0.02 | 0.02 | 0.02 | 0.02 | g |
| $NaMoO_4*2H_2O$ | 0.02 | 0.03 | 0.03 | 0.03 | g |
| $Na_2SeO_3*5H_2O$ | 0.02 | / | / | / | g |
| $Na_2SeO_3$ | / | 0.02 | 0.02 | 0.02 | g |
| $Na_2WO_4*2H_2O$ | 0.02 | 0.02 | 0.02 | 0.02 | g |
| $H_3BO_3$ | / | 0.3 | 0.3 | 0.3 | g |
| $MgSO_4*7H_2O$ | / | 3 | 3 | 3 | g |
| NaCl | / | 1 | 1 | 1 | g |
| $AlK(S_2O_8)*12H_2O$ | / | 0.0199 | 0.0199 | 0.0199 | g |
| Vitamin Solution (Per Liter) | | | | | |
| | **Tanner_Mod** | **Valgepea_Mod** | **Tan_Val** | **Tan_Val + Fe** | |
| pyridoxine-HCl | 10 | 10 | 10 | 10 | mg |
| thiamine-HCl*2H$_2$O | 5 | 50 | 50 | 50 | mg |
| riboflavine | 5 | 50 | 50 | 50 | mg |
| calcium pantothenate | 5 | 50 | 50 | 50 | mg |
| thioctic acid | 5 | 50 | 50 | 50 | mg |
| 4-aminobenzoic acid | 5 | 50 | 50 | 50 | mg |
| nicotinic acid | 5 | 50 | 50 | 50 | mg |
| vitamin B12 | 0.1 | 50 | 50 | 50 | mg |
| biotin | 2 | 20 | 20 | 20 | mg |
| folic acid | 2 | 20 | 20 | 20 | mg |

*2.2. Batch Gas Fermentation Experiments*

Autotrophic growth experiments were carried out in batch conditions using 250 mL serum bottles. The working volume was 50 mL for heterotrophic cultivation and 25 mL for autotrophic cultivation, unless differently specified. The gas mixtures considered in our screening consisted of $CO_2$ and $H_2$, or CO and $CO_2$ or simulated syngas. In the $CO \cdot CO_2$ case, we also considered different gas ratios. Therefore, overall six different gas compositions were tested: $CO_2 \cdot H_2$ in the ratio 1:4 (20% $CO_2$ and 80% $H_2$), $CO \cdot CO_2$ in the ratio 4:1 (80% CO and 20% $CO_2$), $CO \cdot CO_2$ in the ratio 1:1 (50% CO and 50% $CO_2$), $CO \cdot CO_2$ in the ratio 1:4 (20% CO and 80% $CO_2$), pure CO (100% CO) and simulated syngas consisting of CO, $CO_2$, $H_2$, $N_2$ in the ratio 5:2:2:1 (50% CO, 20% $CO_2$, 20% $H_2$ and 10% $N_2$) which was inspired by the gas mixture adopted by Valgepea and co-workers [13]. The above gas mixtures were prepared in situ by using a customized gas station equipped with four Mass Flow Controllers (MFCs) (Burkert, Germany) and a gas mixer (Tecnodelta Srl, Turin, Italy). Serum bottles were pressurized to a final overpressure of 0.75 bar. For each gas fermentation experiment the same inoculation procedure was adopted. Glycerol stocks of *C. autoethanogenum* or *C. ljungdahlii* were reactivated in the "Tanner_mod" medium added with 5 g/L of fructose. Strain adaptation to a specific medium ("Tanner_mod", "Valgepea_mod", "Tan_Val" or "Tan_Val + Fe") was carried out in heterotrophy. A mid-exponential grown culture was used as inoculum for autotrophic pre-culture preparation. The autotrophic pre-culture step made use of the specific gaseous substrate that needed to be evaluated. Gas fermentation experiments were inoculated with the autotrophic pre-culture in mid-late exponential growth phase. Tests were run at least in triplicate, at 37 °C and horizontally (or vertically, when specified) agitated at 100 rpm. Sampling for analysis

of growth, pH and products' profile occurred daily by removing 1 mL of liquid from the bottles, using a sterile syringe. Depending on the experiment, from 10 to 15 sampling points were evaluated. Experiments' duration ranged between 10 and 15 days. Cell growth was constantly monitored by measuring optical density at 600 nm ($OD_{600\,nm}$), using a cell density meter model 40 (Thermo Fisher Scientific, Waltham, MA, USA), the pH of the cultures was monitored via pH meter Accumet® AB150 pH/mV pH meter (Thermo Fisher Scientific, Waltham, MA, USA), and the concentrations of products of interest were measured by high performance liquid chromatography (HPLC).

### 2.3. Calculation of Biomass Concentration

Biomass concentration (gCDW/L) was estimated through two different OD to cell dry weight (CDW) correlation coefficients. A correlation coefficient of 0.21 was employed for *C. autoethanogenum* as previously described by Valgepea and co-workers [13], whereas a correlation coefficient of 0.30 was employed for *C. ljungdahlii* as previously described by Infantes and co-workers [43].

### 2.4. Calculation of Batch Fermentation Metrics

Acetate, ethanol and 2,3-BDO productivities were calculated in grams/liter/hour (g/L/h) by dividing the difference of the titer (g/L) of a product at two consecutive sampling points by the number of hours between the two sampling points. This calculation was carried out for every sampling point in an experiment and only the maximum productivity was reported in the Supplementary Tables. The specific growth rates were calculated in grams of cell dry weight (gCDW)/liter/hour (gCDW/L/h), considering only the exponential growth phases and by dividing the difference of the natural logarithm of the CDW values at two different growth points by the number of hours between the two growth points. Finally, products' ratios were calculated by dividing the titer (g/L) of two different products.

### 2.5. Analytics

Quantification of fructose, acetate, ethanol, R,R-2,3-butanediol, meso-2,3-butanediol and lactate was carried out using a "Dionex UltiMate 3000" HPLC system (Thermo Fisher Scientific, Waltham, MA, USA) coupled with a refractive index detector "RefractorMax 520" (RID) (ERC, Tokyo, Japan) operating at 55 °C and a "Dionex UltiMate 3000" variable wavelength detector (Thermo Fisher Scientific, Waltham, MA, USA) set at 210 nm. The "Metab-AAC BF" series column (length = 300 mm; ID = 7.8 mm) (Isera, Düren, Germany) with a pre-column was eluted isocratically with 9 mM $H_2SO_4$ at a flow rate of 0.6 mL/min and an oven temperature of 40 °C. Slightly acidified water (0.009 M $H_2SO_4$) was employed as the mobile phase. Prior to HPLC analysis, culture samples were centrifuged for 10 min at 10,000 rpm in a MicroCL 17R Centrifuge (Thermo Fisher Scientific, Waltham, MA, USA) and filtered through 0.22 μm filters (Scharlab, Barcelona, Spain).

### 2.6. Data Analysis

Data analysis and graphs were carried out via the GraphPad Prism 9.2.0 (GraphPad, San Diego, CA, USA) software. All data derived from triplicate experiments are shown as mean values ± standard error of the mean (SEM). ANOVA and the follow-up Tukey test were performed in order to compare the means of the distribution of the CDW or of the titer of a certain product among multiple conditions. Multiplicity adjusted Tukey's test *p*-values are shown in relation to each comparison displayed in the plots. Single, paired, and one-tailed t-tests were carried out to compare the means of the distributions of a certain observable obtained in two different conditions.



## 3. Results and Discussion

### 3.1. Cultivation Technique Optimization for the Autotrophic Growth of a Model Acetogen

Prior to the gaseous substrate screening, we tested different cultivation conditions in the attempt to improve the gas to liquid mass transfer and, consequently, the autotrophic *C. ljungdahlii* growth and 2,3-butanediol (2,3-BDO) production [3]. Ensuring that the gas substrate is available for cellular uptake is fundamental in the microbial conversion of gaseous substrates to bioproducts. Gas substrate availability is influenced by the transport of the gas from the gaseous phase to the liquid phase and then to the cell surface [36,44,45]. At the lab-scale, microbial gas fermentation is generally carried out in serum bottles [32,34] and, in this case, the primary factor influencing the gas to liquid mass transfer is the gas to liquid volumetric ratio [37].

Here, we compared three different cultivation methods to enhance the gas to liquid mass transfer in serum bottles. *C. ljungdahlii* was grown on the "Tanner_mod", a modified Tanner medium [27,42], according to the specifications provided in the Materials and Methods section. We adopted the cultivation method consisting of vertically incubated 250 mL serum bottles containing 50 mL of liquid medium [34,46] as the control. The second condition tested foresaw an increase in the gas to liquid volumetric ratio from 4:1 to 9:1. In the third condition tested, serum bottles with a gas to liquid volumetric ratio of 9:1 were placed horizontally to verify whether the increased gas-liquid contact surface could positively influence cellular growth or products' profile. Thus, the three conditions tested can be summarized as follows, (i) vertically incubated bottles with a gas to liquid ratio of 4:1; (ii) vertically incubated bottles with a gas to liquid ratio of 9:1 and (iii) horizontally incubated bottles with a gas to liquid ratio of 9:1. *C. ljungdahlli* was autotrophically cultivated in parallel using each cultivation condition and $CO \cdot CO_2$ 4:1 (80% CO and 20% $CO_2$) as the gaseous substrate. *C. ljungdahlii* growth was found to take advantage of increasing the gas to liquid volumetric ratio but was not influenced by the orientation used during incubation. Indeed, maximal $OD_{600\,nm}$ values of $0.83 \pm 0.02$, $1.46 \pm 0.12$ and $1.39 \pm 0.03$ were obtained using the 4:1 vertical, 9:1 vertical and 9:1 horizontal cultivation conditions, respectively (Figure 1A). As concerns products' formation, the choice of the cultivation method mildly influenced the formation of acetate (Figure S1) and ethanol, where the maximal titer obtained with the 9:1 vertical condition ($0.29 \pm 0.01$ g/L) was 1.52-fold higher than the minimal one, attained using the 4:1 vertical condition ($0.19 \pm 0.00$ g/L) (Figure 1B). Instead, the most pronounced influence of the cultivation mode was observed on 2,3-BDO production. In particular, adopting a gas to liquid ratio of 9:1 and a horizontal orientation of the serum bottles afforded a 2,3-BDO titer of $1.25 \pm 0.04$ g/L, which is 1.26-fold and 5-fold higher than the titers obtained when *C. ljungdahlii* was cultivated with a gas to liquid ratio of 9:1 and vertically incubated serum bottles ($0.99 \pm 0.04$ g/L) or with a gas to liquid ratio of 4:1 and vertically incubated serum bottles ($0.25 \pm 0.01$ g/L), respectively (Figure 1C). These results confirmed the observation made by Lee and co-workers [37] whereby supplying a higher amount of gaseous substrate is necessary in order to obtain 2,3-BDO autotrophic production. Interestingly, also 2,3-BDO productivities and ethanol:2,3-BDO ratio were greatly influenced by the cultivation technique (Tables S1 and S2). In fact, the 2,3-BDO productivity increased by 5.75-fold in the 9:1 vertical configuration ($0.023 \pm 0.002$ g/L/h) compared to the 4:1 vertical one ($0.004 \pm 0.000$ g/L/h). An additional increment in 2,3-BDO productivity was obtained by testing the 9:1 horizontal configuration, where 2,3-BDO productivity was $0.027 \pm 0.001$ g/L/h (Table S1). Correspondingly, the 9:1 horizontal cultivation condition improved the ethanol:2,3-BDO ratio which equated 1:4.90 (Table S2).

It is plausible that 2,3-BDO production is more favored than acetate or ethanol by the increased turbulence created by the agitation in horizontal mode since increasingly reduced products such as 2,3-BDO particularly benefit from an excess supply of energy (e.g., electrons from CO) [47]. Due to the increased 2,3-BDO titer obtained by using a gas to liquid ratio of 9:1 and a horizontal incubation, this cultivation configuration was used as the reference cultivation technique for performing all the subsequent experiments included in this work.

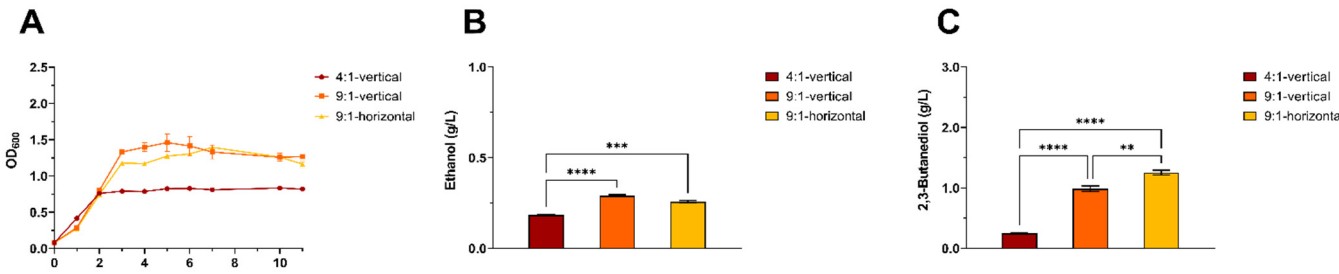

**Figure 1.** Comparative analysis of growth and production profile of *C. ljungdahlii* cultivated in CO · CO$_2$ 4:1 and using different cultivation techniques. Evaluated configurations were vertically incubated bottles with a gas to liquid ratio of 4:1 (rosewood red), vertically incubated bottles with a gas to liquid ratio of 9:1 (Spanish orange), and horizontally incubated bottles with a gas to liquid ratio of 9:1 (Indian yellow), during 250 mL serum bottles batch experiments. (**A**) Growth profile in the three cultivation conditions. (**B**) Ethanol maximum titer, measured at day 5, 6 and 7 for the 4:1-vertical, 9:1-vertical and 9:1-horizontal conditions, respectively. (**C**) 2,3-butanediol maximum titer, measured at day 5 for the 4:1-vertical and 9:1-vertical conditions and at day 7 for the 9:1-horizontal condition. The mean of three independent replicates is plotted for each tested condition. Error bars indicate the standard error of the mean (SEM). ANOVA P: ethanol ($p < 0.0001$); 2,3-BDO ($p < 0.0001$). Tukey's test *p*: ** $p < 0.0021$; *** $p < 0.0002$ and **** $p < 0.0001$.

### 3.2. Screening of Gaseous Substrates for Enhanced 2,3-BDO Production Using Clostridium ljungdhalii and Clostridium autoethanogenum

The gaseous substrate composition alters the autotrophic metabolism and the production profile of a fermentation culture [7,8,10,33,48]. In spite of the breadth of studies focusing on different gaseous mixtures and strains [7,12,14], capitalizing on the reported outcomes is penalized by the scarce comparability between different studies [38]. Here, to overcome this limitation and ensure an accurate comparability of the experimental outcomes, we maintained the same experimental setup (inoculum, temperature, agitation speed, pH, and pressure) and sampling technique in each batch fermentation, carried out to explore the effects of different gas substrates on products' formation. *C. ljungdahlii* and *C. autoethanogenum* were grown on the "Tanner_mod" medium" [27,42]. The three gaseous substrates explored included: (i) CO$_2$ · H$_2$ in the ratio 1:4 (20% CO$_2$ and 80% H$_2$); (ii) CO · CO$_2$ in the ratio 4:1 (80% CO and 20% CO$_2$) and (iii) simulated syngas consisting of CO, CO$_2$, H$_2$, N$_2$ in the ratio 5:2:2:1 (50% CO, 20% CO$_2$, 20% H$_2$ and 10% N$_2$). Gas ratios and percentages will no longer be reported explicitly, except for the CO · CO$_2$ case for which the gas ratio influence was investigated in Section 3.4.

Growth and production metrics corresponding to *C. ljungdahlii* cultures inoculated with a headspace of each of the aforementioned gaseous mixture were comparatively assessed (Figure 2 and Figure S2, Tables S3 and S4). As shown in Figure 2A, the CO$_2$ · H$_2$ gas mixture, allowed to reach the least OD$_{600nm}$, compared to CO · CO$_2$ 4:1 and syngas. Indeed, the maximum OD$_{600nm}$ attained under CO$_2$ · H$_2$ was only $0.53 \pm 0.02$, in accordance with previous observations [8,49]. When looking into the effects of the choice of the gaseous substrate on products' formation, the CO$_2$ · H$_2$ gas mixture was slightly preferable for the formation of ethanol. Indeed, the maximum ethanol titers obtained using CO$_2$ · H$_2$, CO · CO$_2$ 4:1 and syngas as gaseous substrates were $0.29 \pm 0.01$, $0.23 \pm 0.00$ and $0.20 \pm 0.00$ g/L, respectively (Figure 2B). The 2,3-BDO titer was the most sensitive to this choice. In fact, *C. ljungdahlii* produced $1.27 \pm 0.02$ g/L of 2,3-BDO when grown on CO · CO$_2$ 4:1, which is 2.8-fold higher than the titer attained when *C. ljungdahlii* was grown on syngas, whereas no 2,3-BDO production was detected when CO$_2$ · H$_2$ was employed (Figure 2C). Our results neatly indicate that the CO$_2$ · H$_2$ gas mixture is not appropriate for 2,3-BDO production, confirming previous observations [7,8], and point at CO · CO$_2$ 4:1 as the preferable gaseous substrate for 2,3-BDO autotrophic production. Notably, growing *C. ljungdahlii* on CO · CO$_2$ 4:1 allows an ethanol:2,3-BDO ratio of 1:6.03 (Table S4). The favorable 2,3-BDO titer obtained growing *C. ljungdahlii* on CO · CO$_2$ 4:1 depends on the energetic favorable reduction power of CO, which enhances the ATP yield of 2,3-BDO pro-

duction, compared to $H_2$ [10,36]. In autotrophically grown *C. ljungdahlii*, all the produced pyruvate, a key precursor for 2,3-BDO production, is derived from acetyl-CoA through a single step catalyzed by a pyruvate:ferredoxin oxidoreductase (PFOR). As acetyl-CoA is a $C_2$ compound and pyruvate is a $C_3$ compound, a molecule of $CO_2$ needs to be incorporated. The energy for this reaction is provided by reduced ferredoxin [50]. The usage of the gas mixture, consisting of both CO and $CO_2$, can favor 2,3-BDO production by increasing the levels of both $CO_2$ and reduced ferredoxin, the reactants in the reaction catalyzed by the PFOR. Indeed, this gas mixture can directly provide $CO_2$ and, owing to the presence of CO, can increase the level of reduced ferredoxin by CO oxidation via the ferredoxin-dependent carbon monoxide dehydrogenase (CODH). The 2,3-BDO production allows *C. ljungdahlii* to neutralize pyruvic acid whose excess could threaten the transmembrane proton gradient, which is required for ATP formation [50].

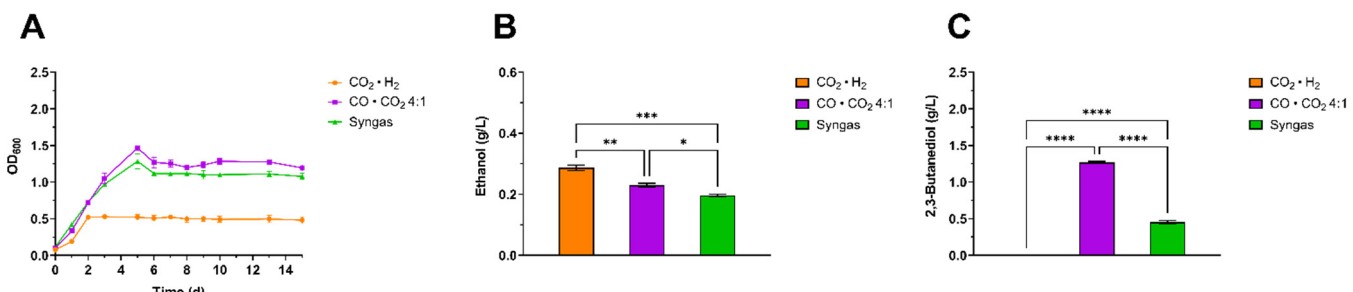

**Figure 2.** Comparative analysis of growth and production profile of *C. ljungdahlii* using different gaseous substrates. The headspace of 250 mL serum bottles was filled with either $CO_2 \cdot H_2$ (orange), $CO \cdot CO_2$ 4:1 (violet) or syngas (CO, $CO_2$, $H_2$, $N_2$ 5:2:2:1) (green). (**A**) Growth profile characterization with the three gas mixtures. (**B**) Ethanol maximum titer, measured at day 15 for the three conditions tested. (**C**) 2,3-butanediol maximum titer, measured at day 5 and 3 for the $CO_2 \cdot H_2$ and $CO \cdot CO_2$ 4:1 conditions, respectively. The mean of three independent replicates is plotted for each tested condition. Error bars indicate the standard error of the mean (SEM). ANOVA *p*: ethanol ($p = 0.0002$); 2,3-BDO ($p < 0.0001$). Tukey's test *p*: * $p < 0.0332$; ** $p < 0.0021$; *** $p < 0.0002$ and **** $p < 0.0001$.

As for *C. ljungdahlii*, growth and production specificities were assessed also in *C. autoethanogenum* grown on each gas mixture, using identical cultivation and sampling techniques to enable fair comparison of the outcomes (Figure 3 and Figure S3). We confirmed that $CO \cdot CO_2$ 4:1 afforded the highest $OD_{600nm}$, as shown in Figure 3A and, as a side result, that $CO_2 \cdot H_2$ afforded the highest ethanol titer (Figure 3B). The highest 2,3-BDO titer was achieved by the $CO \cdot CO_2$ 4:1 gas substrate, when compared to the other two gas substrates (Figure 3C). Indeed, the 2,3-BDO titer obtained using $CO \cdot CO_2$ 4:1 ($0.37 \pm 0.03$ g/L) was found statistically significantly higher than that obtained using syngas ($0.20 \pm 0.01$ g/L), whereas $CO_2 \cdot H_2$ led to no 2,3-BDO. Collectively, the outcomes of the gas substrates' screening, conducted in full similarity using either *C. ljungdahlii* or *C. autoethanogenum*, allowed us to establish $CO \cdot CO_2$ 4:1 as the preferable substrate for 2,3-BDO production.

Upon identification of the preferable gas substrate for producing 2,3-BDO, we selected the appropriate acetogenic strain by comparing the maximal titer and productivity achieved by *C. ljungdahlii* and *C. autoethanogenum* grown on $CO \cdot CO_2$ 4:1. This comparison revealed an interesting product specificity of these strains. *C. autoethanogenum* is preferable to *C. ljungdahlii* for ethanol production, by affording a 5.72-fold higher titer than *C. ljungdahlii* under the $CO_2 \cdot H_2$ condition (Figure 4A). More importantly, we found that *C. ljungdahlii* is preferable to *C. autoethanogenum* for 2,3-BDO production. Indeed, growing *C. ljungdahlii* on $CO \cdot CO_2$ 4:1 afforded a 3.43-fold higher titer compared to *C. autoethanogenum* (Figure 4B), grown on the same gas substrate. It is laterally noted that *C. ljungdahlii* was found to be preferential for 2,3-BDO production also when using syngas (Figure 4C). Moreover, the maximum 2,3-BDO productivity in *C. ljungdahlii* ($0.024 \pm 0.001$ g/L/h) was 2.66-fold higher than that in *C. autoethanogenum* ($0.009 \pm 0.001$ g/L/h), when $CO \cdot CO_2$ 4:1 was used as gas substrate (Table S3). According to these findings, we set out to select *C. ljungdahlii*

and CO · CO₂ 4:1 as the preferable strain and gas substrate to be used in the subsequent medium screening study to enhance 2,3-BDO production.

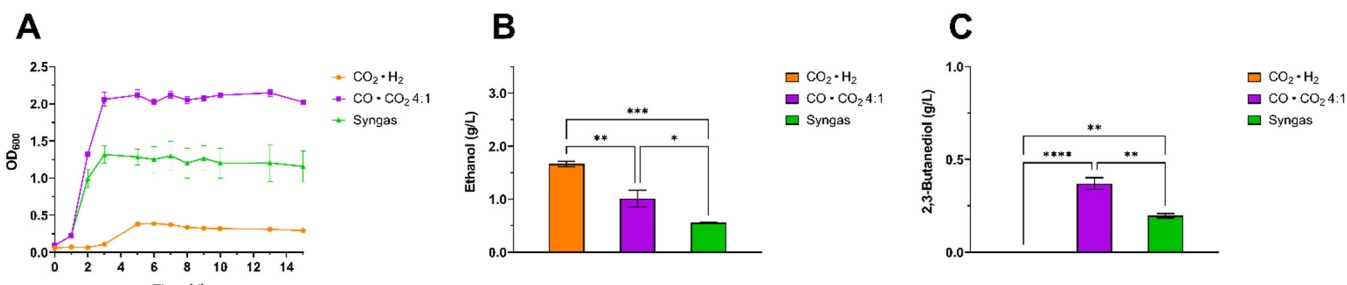

**Figure 3.** Comparative analysis of growth and production profile of *C. autoethanogenum* using different gaseous substrates. The headspace of 250 mL serum bottles was filled with either CO₂ · H₂ (orange), CO · CO₂ 4:1 (violet) or syngas (CO, CO₂, H₂, N₂ 5:2:2:1) (green). (**A**) Growth profile characterization with the three gas mixtures. (**B**) Ethanol maximum titer, measured at day 8 for the CO₂ · H₂ condition and at day 15 for the CO · CO₂ 4:1 and syngas conditions. (**C**) 2,3-butanediol maximum titer, measured at day 3 and 5 for the CO · CO₂ 4:1 and syngas conditions, respectively. The mean of three independent replicates is plotted for each tested condition. Error bars indicate the standard error of the mean (SEM). ANOVA *p*: ethanol ($p = 0.0006$); 2,3-BDO ($p < 0.0001$). Tukey's test *p*: * $p < 0.0332$; ** $p < 0.0021$; *** $p < 0.0002$ and **** $p < 0.0001$.

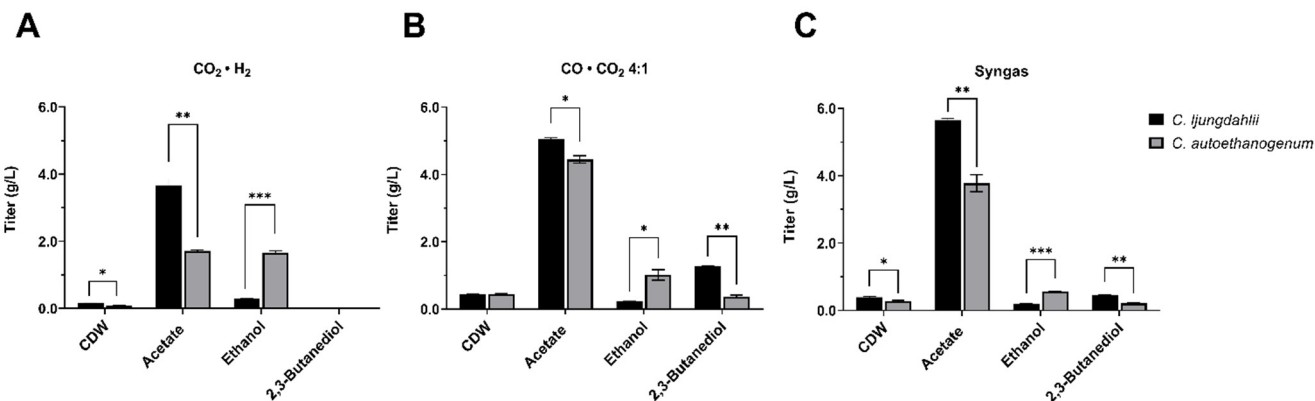

**Figure 4.** Comparative analysis of growth and production profiles of *C. ljungdahlii* (black) and *C. autoethanogenum* (grey) using different gaseous substrates. The headspace of 250 mL serum bottles was filled with either (**A**) CO₂ · H₂, (**B**) CO · CO₂ 4:1 or (**C**) syngas (CO, CO₂, H₂, N₂ 5:2:2:1). Maximum values of cell dry weight (CDW), acetate, ethanol and 2,3-butanediol are reported. The mean of three independent replicates for each gaseous substrate is plotted. Error bars indicate the standard error of the mean (SEM). For each gaseous substrate tested, shown are the single, paired one-tailed t-tests, corresponding to statistically significant differences of the observable (CDW, titer of each product) between *C. ljungdahlii* and *C. autoethanogenum* (* $p < 0.0332$; ** $p < 0.0021$; *** $p < 0.0002$).

### 3.3. Influence of Cultivation Medium on 2,3-BDO Production from CO · CO₂ using Clostridium ljungdahlii

In order to determine the preferable medium composition for 2,3-BDO production, we screened out three additional media featuring different concentrations of trace metals, mineral elements and vitamins, as shown in Table 2.

Although previous studies have explored medium definition to support ethanol production [37,40,41,51], the literature lacks a systematic characterization of the medium composition favoring 2,3-BDO acetogenic production. In full similarity to the screening on gaseous substrates, to enable an appropriate comparison of the results, we carried out the media screening by using the same inoculation, cultivation condition, sampling technique, gas substrate (CO · CO₂ 4:1) and biocatalyst (*C. ljungdahlii*), previously selected.

**Table 2.** Summary of the final concentration of the main elements in the media tested in this study. Final concentrations of mineral elements (mM), trace elements (metal) (µM), and vitamins (µM) are reported in the table.

| Mineral Solution (mM) | | | | |
|---|---|---|---|---|
| | **Tanner_Mod** | **Valgepea_Mod** | **Tan_Val** | **Tan_Val + Fe** |
| NaCl | 34.22 | 3.59 | 34.39 | 34.39 |
| NH$_4$Cl | 46.74 | 46.74 | 46.74 | 46.74 |
| KCl | 3.35 | 6.71 | 3.35 | 3.35 |
| PO$_4$ | 1.84 | 16.99 | 1.84 | 1.84 |
| Mg | 2.03 | 2.58 | 2.15 | 2.15 |
| Ca | 0.68 | 0.90 | 0.68 | 0.68 |
| **Trace Element (Metal) Solution (µM)** | | | | |
| | **Tanner_Mod** | **Valgepea_Mod** | **Tan_Val** | **Tan_Val + Fe** |
| Mn | 59.17 | 29.58 | 29.58 | 29.58 |
| Fe | 20.40 | 86.89 | 23.99 | 86.89 |
| Co | 8.41 | 8.41 | 8.41 | 8.41 |
| Zn | 0.03 | 6.96 | 6.96 | 6.96 |
| Cu | 1.17 | 1.17 | 1.17 | 1.17 |
| Ni | 0.84 | 0.84 | 0.84 | 0.84 |
| Mo | 0.83 | 1.24 | 1.24 | 1.24 |
| Se | 0.76 | 1.16 | 1.16 | 1.16 |
| W | 0.61 | 0.61 | 0.61 | 0.61 |
| B | / | 48.52 | 48.52 | 48.52 |
| Al | / | 0.84 | 0.84 | 0.84 |
| **Vitamin Solution (µM)** | | | | |
| | **Tanner_Mod** | **Valgepea_Mod** | **Tan_Val** | **Tan_Val + Fe** |
| pyridoxine | 486.29 | 486.29 | 486.29 | 486.29 |
| thiamine | 148.24 | 1482.36 | 1482.36 | 1482.36 |
| riboflavine | 132.85 | 1328.52 | 1328.52 | 1328.52 |
| pantothenate | 104.93 | 1049.32 | 1049.32 | 1049.32 |
| thioctic acid | 242.33 | 2423.30 | 2423.30 | 2423.30 |
| 4-aminobenzoic acid | 364.59 | 3645.91 | 3645.91 | 3645.91 |
| nicotinic acid | 406.14 | 4061.41 | 4061.41 | 4061.41 |
| vitamin B12 | 0.74 | 368.63 | 368.63 | 368.63 |
| biotin | 81.86 | 818.63 | 818.63 | 818.63 |
| folic acid | 45.31 | 453.10 | 453.10 | 453.10 |

In addition to the "Tanner_mod" medium, our screening included three alternative media that are referred to as "Valgepea_mod", "Tan_Val" and "Tan_Val + Fe" hereafter and that are detailed in the Materials and Methods section. The "Valgepea_mod" medium, slightly modified from Valgepea and co-workers [13], differs from the "Tanner_mod" one for the concentrations of mineral elements, trace metals, vitamins and Iron, as shown in Table 2. The "Tan_Val" medium has been conceived in this work and foresees the mineral elements of the "Tanner_mod" medium and the trace metals and vitamins of the "Valgepea-mod" medium. Subsequently, in order to investigate the Iron influence on 2,3-BDO production, we formulated the "Tan_Val + Fe" medium that is identical to the "Tan_Val" one but foresees the same Iron concentration of the "Valgepea_mod" medium–86.89 µM (Table 2). Yeast extract (YE) was removed from the formulation of all the media different from the "Tanner_mod" one, which included 0.5 g/L of YE, in the drive to reduce the costs, to favor the formation of non-growth-coupled products and to disambiguate the products' profile interpretation [40,41,52].

Growth of *C. ljungdahlii* on CO · CO$_2$ 4:1 using the "Tanner_mod" medium did not show the lag phase, compared to other conditions, probably due to the presence of the optimal YE concentration [41,53]. The "Tan_Val" medium afforded a maximum OD$_{600nm}$ (1.56 ± 0.04 at day 6) similar to the one reached using the "Tanner_mod" medium

(1.47 ± 0.02 at day 5) and led to the highest specific growth rate (0.0346 ± 0.0007 gCDW/L/h, Table S5), whereas growth on the "Valgepea_mod" and on the "Tan_Val + Fe" media, led to lower maximum $OD_{600nm}$ values (1.10 ± 0.13 at day 7 and 1.33 ± 0.04 at day 10, respectively), as shown in Figure 5A. We attributed the negative effect of the latter media on growth to the higher Iron concentration, which distinguishes the "Valgepea_mod" and "Tan_Val + Fe" media from the "Tanner_mod" and "Tan_Val" ones. In fact, the "Tan_Val" and the "Tanner_mod" media have similar Iron concentration (23.99 µM and 20.40 µM, respectively), while the "Valgepea_mod" and "Tan_Val + Fe" media are characterized by the same Iron concentration (86.89 µM) which is 4.25-fold higher than that of the "Tanner_mod" medium and 3.62-fold higher than that of the "Tan_Val" medium (Table 2).

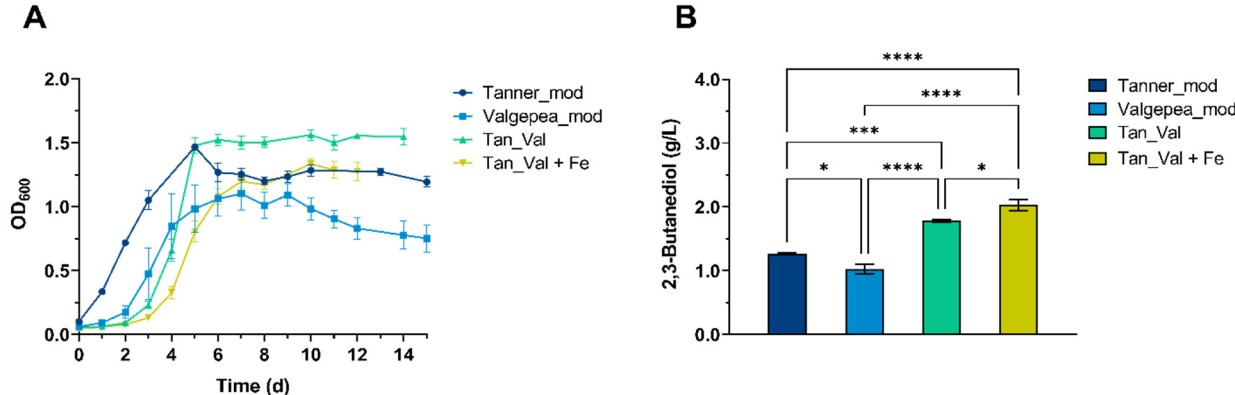

**Figure 5.** Comparative analysis of growth and 2.3-butanediol production of *C. ljungdahlii* grown in CO · CO$_2$ 4:1 and in different media. "Tanner_mod" (Yale blue); "Valgepea_mod" (Munsell blue); "Tan_Val" (Malachite green) and "Tan_Val + Fe" (Chartreuse yellow) media were employed during 250 mL serum bottles batch experiments. (**A**) Growth profile with the four cultivation media. (**B**) Maximum 2,3-butanediol titer, measured at day 5, 10, 5 and 10 for the "Tanner_mod", "Valgepea_mod", "Tan_Val" and "Tan_val + Fe" conditions, respectively. The mean of three independent replicates is plotted for each cultivation medium. Error bars indicate the standard error of the mean (SEM). ANOVA P: 2,3-BDO ($p < 0.0001$). Tukey's test $p$: * $p < 0.0332$; *** $p < 0.0002$ and **** $p < 0.0001$.

Medium formulation drastically influenced 2,3-BDO titer and productivity. In fact, 1.27 ± 0.02 g/L, 1.02 ± 0.08 g/L, 1.78 ± 0.03 g/L and 2.03 ± 0.05 g/L of 2,3-BDO were obtained by using the "Tanner_mod", "Valgepea_mod", "Tan_Val" and "Tan_Val + Fe" media after 5, 10, 5 and 10 days from the beginning of the test, respectively (Figure 5B). Whereas 2,3-BDO titers in the "Valgepea_mod" and "Tanner_mod" media were similar, 2,3-BDO productivity in the "Valgepea_mod" medium (0.012 ± 0.005 g/L/h) was 2-fold lower than using the "Tanner_mod" medium (0.024 ± 0.001 g/L/h) (Table S5). In addition to different mineral elements' concentrations, the "Valgepea_mod" medium contains 200-, 1.5-, 1.5- and 4.25-fold higher concentrations of Zinc, Molybdenum, Selenium and Iron, respectively, compared to the "Tanner_mod" medium (Table 2). A higher concentration of Zinc, Molybdenum, Selenium and Iron has been previously reported to increase cell growth and ethanol production in *C. ragsdalei* grown on CO · CO$_2$ [51]. Although a similar positive effect was not observed in our study on 2,3-BDO productivity and titer, our results do not rule out the possibility that increasing trace metals in the growth medium could be beneficial also to the 2,3-BDO production since the absolute amounts and relative increase of the trace metals in our study are neatly lower than in Saxena and Tanner [51]. Metal elements are essential to support the activity of metalloenzymes in the Wood-Ljungdahl pathway [54,55]. Moreover, the concentration of most constituents of the vitamin solution of the "Valgepea_mod" medium was increased by 10-fold, with the exception of vitamin B12, which was barely detectable in the "Tanner_mod" medium and was augmented by around 500-fold in the "Valgepea_mod" medium (Table 2). Although some studies did not provide evidence that manipulating vitamin concentrations could affect either the growth or the production profile of *C. autoethanogenum* [40,53], it was

shown that increasing the concentration of specific vitamins, such as pantothenic acid and biotin, above the cellular requirements of an acetogen grown on a CO-rich gas mixture led to an increase in 2,3-BDO production [56]. Furthermore, a role in anaerobic $CO_2$ fixation was described for vitamin B12, a cofactor for methyl group transfer reactions in the Wood-Ljungdahl pathway [57], whose concentration was substantially increased in the "Valgepea_mod" medium. Therefore, we speculated that it was the mineral elements defined in the "Valgepea_mod" medium (Table 2) which could exert a negative effect on 2,3-BDO production, despite the higher Zinc concentration present in the "Valgepea_mod" medium, compared with the "Tanner_mod" one.

Therefore, we tested the "Tan_Val" medium, which combines the trace metals' and vitamins' formulation of the "Valgepea_mod" medium with the mineral elements' concentration used in the "Tanner_mod" medium (Table 2). We managed to enhance both 2,3-BDO titer (1.78 ± 0.03 g/L) and 2,3-BDO productivity (0.038 ± 0.005 g/L/h), which were found 1.4- and 1.58-fold higher, respectively, when compared to the "Tanner_mod" medium (Figure 5B, Table S5). The improvement, obtained using the "Tan_Val" medium, could be attributable to the effects exerted, in particular, by the combination of increased concentrations of trace metals and vitamins—derived from the "Valgepea_mod" medium—with the mineral element concentrations derived from the "Tanner_mod" medium. Zinc was one of the trace metals whose concentration mostly changed between the "Tanner_mod" and "Tan_Val" media with a 200-fold increase (Table 2). The positive effect observed could be plausible due to the fact that Zinc is a fundamental metal for acetogenic alcohol dehydrogenase enzymes [58]. In particular, the butanediol dehydrogenase (CLJU_c23220), which is the enzyme responsible for the conversion of acetoin into 2,3-BDO, is a Zinc-dependent enzyme [26]. The beneficial effect of replacing the "Valgepea_mod" mineral solution with the "Tanner_mod" one was confirmed when we compared the 2,3-BDO production obtained using the "Valgepea_mod" and "Tan_Val" media, which differ only for the mineral elements' and Iron concentrations (Table 2). Here, the different concentration of mineral elements present in the "Tan_Val" medium, relative to the "Valgepea_mod" one, corresponded to a 1.74-fold increase in 2,3-BDO titer and to a 3.16-fold increase in 2,3-BDO productivity (Figure 5B).

Finally, we tested 2,3-BDO production in the "Tan_Val + Fe" medium which led to 2.03 ± 0.05 g/L of 2,3-BDO. The titer achieved here is unprecedented on the basis of the titers recorded in previous batch experiments on serum bottles level. Indeed, the highest titer by a pure culture was of 1 g/L but was obtained by growing *C. autoethanogenum* culture on syngas supplemented with fructose [14]. The titer obtained in our study was found to be similar to the titer previously obtained by a syntrophic cultivation of *C. acetobutylicum* and *C. ljungdahlii* which was equal to 1.98 g/L, but in this case, glucose was used as an additional carbon source [59].

The "Tan_Val + Fe" medium differs only for the concentration of mineral elements from the "Valgepea_mod" medium and differs only for the Iron concentration from the "Tan_Val" medium (Table 2). By comparing 2,3-BDO production titers obtained in the "Tan_Val + Fe" and "Valgepea_mod" media, we confirmed the positive influence of the mineral elements on 2,3-BDO production (Figure 5B). Moreover, by comparing 2,3-BDO production titers obtained in the "Tan_Val" and "Tan_Val + Fe" media, we found that increasing Iron concentration could have a slightly positive effect on 2,3-BDO production (Figure 5B). In fact, Iron serves as essential metal cofactor for the Iron-Sulphur proteins, which are related to the electron transfer pathway [18], the methyl-branch [60] and carbonyl-branch [55,61] of the Wood-Ljungdahl pathway. Moreover, Iron was found to be a limiting element for acetogens' growth and productivity [41,62] and also an important coenzyme of alcohol dehydrogenase [63].

The results summarized as follows let us to conclude that trace elements and vitamins generally have only a minor effect on 2,3-BDO production while mineral elements, Zinc and Iron exert a major positive influence on 2,3-BDO titer and productivity. The higher titer and productivity achieved by the "Tan_Val + Fe" compared to the "Valgepea_mod" medium de-

pends on the replacement of the "Valgepea_mod" mineral element concentrations with the "Tanner_mod" ones. The "Tanner_mod" mineral solution features 9.23- and 2-fold lower $PO_4$ and KCl concentrations and 9.58-fold higher NaCl compared to the "Valgepea_mod" medium whereas Ca concentrations were similar. The higher titer achieved by the "Tan_Val + Fe" medium compared to the "Tan_Val" medium is due to the increased Fe supplementation. Finally, the similar titers of "Valgepea_mod" and "Tanner_mod" together with the lower productivity of the "Valgepea_mod" medium compared to the "Tanner_mod" one let us think that positive effect of increased Zn in the "Valgepea_mod" medium was reduced by the "Valgepea_mod" mineral element concentrations, which was found to be the only difference responsible of the lower performance of the "Valgepea_mod" medium with respect to the "Tan_Val + Fe" one. It is noted that the "Valgepea_mod" medium was specifically formulated to boost ethanol production of *C. autoethanogenum* using syngas [13], while we tested its influence on 2,3-BDO production in *C. ljungdahlii* grown on $CO \cdot CO_2$. Taken together, our results highlight the importance of undertaking studies on medium composition influence on 2,3-BDO production, as those already performed by Saxena and Tanner [51] and by Gao and co-workers [41] for ethanol production using *C. ragsdalei* grown on $CO \cdot CO_2$.

### 3.4. Characterization of the Effects of Different CO to $CO_2$ Ratios on 2,3-BDO Production by Clostridium ljungdahlii

We investigated the growth and metabolites' profile of *C. ljungdahlii* grown in the " Tan_Val" medium and by using four $CO \cdot CO_2$ gas mixtures, differing in the CO to $CO_2$ gas ratios: pure CO, $CO \cdot CO_2$ 4:1 (80% CO and 20% $CO_2$), $CO \cdot CO_2$ 1:1 (50% CO and 50% $CO_2$) and $CO \cdot CO_2$ 1:4 (20% CO and 80% $CO_2$) (Figure 6 and Figure S5, Tables S7 and S8).

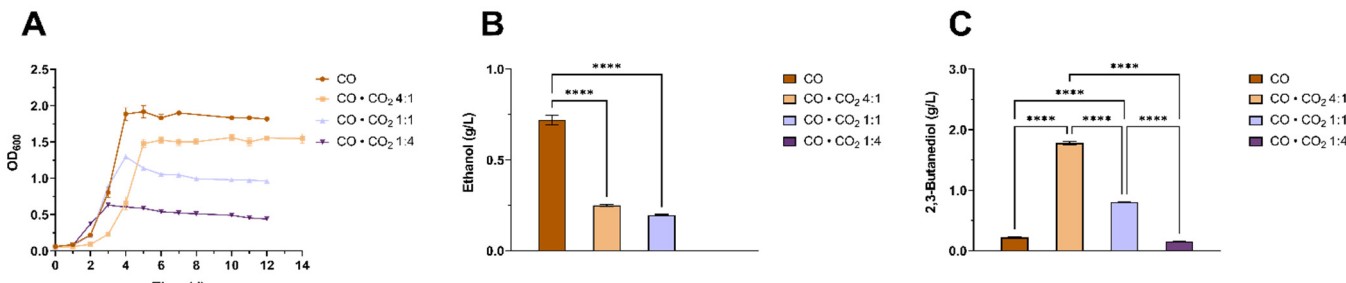

**Figure 6.** Comparative analysis of growth and production profile of *C. ljungdahlii* using different CO:$CO_2$ gas ratios. The headspace of 250 mL serum bottles was filled with either pure CO (brown) $CO \cdot CO_2$ 4:1 (Sandy tan), $CO \cdot CO_2$ 1:1 (Lavender blue) or $CO \cdot CO_2$ 1:4 (dark purple). (**A**) Growth profiles with the four gas mixtures. (**B**) Ethanol maximum titer, measured at day 10, 14 and 4 for the pure CO, $CO \cdot CO_2$ 4:1 and $CO \cdot CO_2$ 1:1 conditions, respectively. (**C**) 2,3-butanediol maximum titer, measured at day 6, 5, 4 and 3 for the pure CO, $CO \cdot CO_2$ 4:1, $CO \cdot CO_2$ 1:1 and $CO \cdot CO_2$ 1:4 conditions, respectively. The mean of three independent replicates is plotted for each tested condition. Error bars indicate the standard error of the mean (SEM). ANOVA P: ethanol ($p < 0.0001$); 2,3-BDO ($p < 0.0001$). Tukey's test $p$: **** $p < 0.0001$.

*C. ljungdahlii* growth on pure CO gas substrate outperformed that on $CO \cdot CO_2$. In fact, *C. ljungdahlii* reached a maximal $OD_{600nm}$ of $1.92 \pm 0.08$, $1.56 \pm 0.04$, $1.30 \pm 0.02$ and $0.63 \pm 0.00$ using pure CO, $CO \cdot CO_2$ 4:1; $CO \cdot CO_2$ 1:1 and $CO \cdot CO_2$ 1:4, respectively (Figure 6A). As expected, by reducing the amount of CO in the gaseous substrate, biomass and acetate, which is a growth-related metabolite, decreased (Figure 6 and Figure S5). Ethanol titers were negatively affected by increasing the percentage of $CO_2$ at the expense of the CO present in the gaseous substrate. By comparing 2,3-BDO titers obtained in the pure CO and $CO \cdot CO_2$ 4:1 conditions, it is interesting to note that the presence of $CO_2$ positively influenced 2,3-BDO production at the expense of ethanol production. In fact, growing *C. ljungdahlii* on pure CO produced only $0.22 \pm 0.01$ g/L of 2,3-BDO whereas the 2,3-BDO titer increased up to $1.78 \pm 0.03$ g/L when *C. ljungdahlii* was grown on $CO \cdot CO_2$ 4:1 (Figure 6B,C). The utilization of a gas mixture consisting of both CO and $CO_2$

can favor 2,3-BDO production since it directly increases the level of $CO_2$ and indirectly, via the CODH-catalyzed reaction, the levels of reduced ferredoxin, the two main reactants of the PFOR reaction [50]. Nevertheless, further increasing the amount of $CO_2$ with respect to CO was revealed to be counterproductive. Indeed, $0.80 \pm 0.01$ g/L and $0.16 \pm 0.00$ g/L of 2,3-BDO were obtained on CO · $CO_2$ 1:1 and CO · $CO_2$ 1:4, respectively, corroborating the indication that the CO presence is a limiting element for 2,3-BDO production (Figure 6C). Indeed, providing sufficient levels of CO allows 2,3-BDO to act as an electron acceptor to relieve the microbial cell of excess of reducing power, in the form of NAD(P)H, thus restoring a favorable NAD(P):NAD(P)H equilibrium. In fact, it has been previously shown that where CO is supplied at sufficient levels, 2,3-BDO dehydrogenase expression is upregulated [64].

## 4. Conclusions

Our study presented a multifaceted screening campaign which, impinging on the selection of a suitable catalyst, gaseous substrate and medium composition, significantly enhanced 2,3-BDO production in batch experiments using serum bottles. The preferable configuration identified *C. ljungdahlii* as biocatalyst and CO · $CO_2$ 4:1 as gas substrate. Our medium screening unveiled that the "Tan_Val" medium outperformed the remaining ones with respect to 2,3-BDO productivity ($0.038 \pm 0.005$ g/L/h) and that the "Tan_Val + Fe" medium ensured the highest 2,3-BDO titer ($2.03 \pm 0.05$ g/L). We identified in the co-presence of optimized mineral elements, Zinc and Iron the major component to boost 2,3-BDO titer and productivity. Moreover, the screening of various CO · $CO_2$ gaseous substrates, characterized by different gas ratios, revealed that $CO_2$ addition to CO fermentation of *C. ljungdahlii* caused a metabolic shift from ethanol to 2,3-BDO production and that $CO_2$ addition over the 20% drastically reduced 2,3-BDO titer. Key to the achievement of enhanced 2,3-BDO titer was the adoption of a rigorous comparative framework preceded by the selection of an improved cultivation configuration (horizontally incubated serum bottles with a gas to liquid ratio of 9:1).

The 2,3-BDO titer of $2.03 \pm 0.05$ g/L, which was achieved in this work by growing *C. ljungdahlii* on CO · $CO_2$ 4:1 and the "Tan_Val + Fe" medium, is superior to any 2,3-BDO titer previously observed in batch fermentation experiments carried out in serum bottles [26,32,36,37]. Similar or higher titers have been previously achieved only in batch [65] or fed-batch fermentation experiments carried out in reactor environments that ensure superior process control [7–9]. A similar 2,3-BDO titer was indeed obtained in a fed-batch fermentation in which a CO · $CO_2$ gas substrate was constantly supplied for 276 h to a *C. ljungdahlii* culture hosted in a bioreactor (working volume, 2.5 L) where gas pressure and pH were constantly controlled at 1 bar and 6.0, respectively [9]. 2,3-BDO titers substantially higher than the value attained in the present study were observed in Zhu and co-workers [7]. In this case, *C. ljungdahlii* was grown in CO · $CO_2$ in the presence of yeast extract in a bioreactor (working volume, 2.5 L) with pH and supplied gas pressure controlled at 6 and 1 bar, respectively, and the 2,3-BDO titer increased up to $16.94 \pm 0.36$ g/L.

Future development will involve further medium optimization by evaluating, for instance, the effects of supplementing specific amino acids to the growth medium on 2,3-BDO production performances [37]. However, we believe that a substantial turnaround in the development of the 2,3-BDO production process will be the optimization of the gaseous conversion into 2,3-BDO in fermentations carried out in gas and liquid continuous configuration. In this case, further medium and gas substrates' optimization could be profitably combined with the optimization of equally relevant fermentation aspects, such as the pH control. Indeed, the pH of the broth affects the energy conservation system which relies on the establishment of a proton-dependent transmembrane ion gradient through the ATPase/Rnf system [7]. In fact, the highest titers of 2,3-BDO, present in literature, were obtained when a constant pH control was applied [7,9]. Furthermore, it is useful to study the effects of different pH profiles on the 2,3-BDO formation. For instance, cyclic pH shifts between a high level of 5.75 and a lower value of 4.75 were carried out in a bioreactor,

continuously fed with carbon monoxide, for continuous ethanol and 2,3-BDO production, resulting in an overall concentration of 1.62 g/L of 2,3-BDO [66].

A yet further parameter that could be controlled, in experiments performed using bioreactors, is the flow rate with which the gas substrate is supplied. Indeed, a *C. autoethanogenum* fermentation carried out in continuous stirred tank reactor prepared at pH 5.5 has previously shown that 2,3-BDO productivity increased significantly from < 0.1 g/L/day to 1.2 g/L/day when the gas flow rate was increased in a such way that specific CO uptake increased from 0.3 to 0.6 mmol/g/min [64]. Finally, foreseeing cell recycling in the configuration of a continuous process for 2,3-BDO production could be beneficial. Aside from the possibility of achieving high cell density cultivation, cell retention could favor 2,3-BDO production since the age of the microbial cells and the growth phase of the microbial culture were found to have an effect on the amount of 2,3-BDO produced. Indeed, 2,3-BDO productivity and 2,3-BDO:ethanol ratio was found to increase with increased average cellular age [23].

In summary, our study enabled the generation of a notable dataset providing a potential guide for continuous gas fermentation processes development for 2,3-BDO production from $CO \cdot CO_2$ fermentation by *C. ljungdahlii*.

**Supplementary Materials:** The following are available online at https://www.mdpi.com/article/10.3390/fermentation7040264/s1, Figure S1: Comparative analysis of acetate production by *C. ljungdahlii* cultivated in $CO \cdot CO_2$ 4:1 and using different cultivation techniques, Figure S2: Comparative analysis of acetate production by *C. ljungdahlii* using different gaseous substrates, Figure S3: Comparative analysis of acetate production by *C. autoethanogenum* using different gaseous substrates, Figure S4: Comparative analysis of acetate and ethanol production of *C. ljungdahlii* grown in $CO \cdot CO_2$ 4:1 and different media, Figure S5: Comparative analysis of acetate production of *C. ljungdahlii* using different $CO:CO_2$ gas ratios, Table S1: Specific growth rate, acetate, ethanol and 2,3-BDO productivities of *C. ljungdahlii* grown in $CO \cdot CO_2$ 4:1 using three different cultivation conditions, Table S2: Product ratios of *C. ljungdahlii* grown in $CO \cdot CO_2$ 4:1 using three different cultivation conditions, Table S3: Specific growth rate, acetate, ethanol and 2,3-BDO productivities of *C. ljungdahlii* and *C. autoethanogenum* grown using three different gaseous substrates, Table S4: Product ratios of *C. ljungdahlii* and *C. autoethanogenum* grown using three different gaseous substrates, Table S5: Specific growth rate, acetate, ethanol and 2,3-BDO productivities of *C. ljungdahlii* grown using four different cultivation media, Table S6: Product ratios of *C. ljungdahlii* grown using four different cultivation media, Table S7: Specific growth rate, acetate, ethanol and 2,3-BDO productivities of *C. ljungdahlii* grown using four different CO-based gaseous substrates, Table S8: Product ratios of *C. ljungdahlii* grown using four different CO-based gaseous substrates.

**Author Contributions:** Conceptualization, L.R. and V.A.; methodology, L.R.; software, L.R.; validation, L.R. and A.R.; formal analysis, L.R. and A.R.; investigation, L.R. and A.R.; resources, V.A. and A.R.; data curation, L.R.; writing—original draft preparation, L.R. and A.R.; writing—review and editing, L.R., V.A., D.F. and A.R.; visualization, L.R. and A.R.; supervision, A.R. and D.F.; project administration, L.R. and A.R.; funding acquisition, A.R. All authors have read and agreed to the published version of the manuscript.

**Funding:** This work was supported by the program P.O.R FESR 2014/2020—Action I.1b.2.2-Bioeconomy Technological Platform—under Grant 333-30 (project PRIME "Processi e pRodotti Innovativi di Chimica vErde").

**Institutional Review Board Statement:** Not applicable.

**Informed Consent Statement:** Not applicable.

**Data Availability Statement:** Data are contained with the article and Supplementary Materials.

**Acknowledgments:** We thank the master thesis student Giacomo Antonicelli for the useful help provided during the investigation and data acquisition processes.

**Conflicts of Interest:** The authors declare that the research was conducted in the absence of any commercial or financial relationships that could be construed as a potential conflict of interest.

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
