# Peer review of "Screening of Gas Substrate and Medium Effects on 2,3-Butanediol Production with C. ljungdahlii and C. autoethanogenum Aided by Improved Autotrophic Cultivation Technique"

_fermentation, doi:10.3390/fermentation7040264_

Round 1
Reviewer 1 Report
The manuscript entitled “Screening of gas substrate and medium effects on 2,3-butane-diol production with C. ljungdahlii and C. autoethanogenum aided by optimized autotrophic cultivation technique” presents very important results related to the production of a chemical platform, 2,3-BDO, by bacterial species. This is a very attractive process because it can use waste gases and produce an important chemical in a biorefinery context. The manuscript is well written, and conclusions are supported by data. There are important results related to medium composition and gas composition for 2,3-BDO production.
Some points should be addressed before publication.
Materials and Methods
MES, MESNA, PETC: abbreviations should be specified in the text
Table 1 should contain all media composition; The description of media composition in section 2.1 is confusing.
Section 2.2: Sampling technique should be specified. The number of samplings per batch also. It influences gas availability. I has also an effect in the calculation of productivity and evaluation if maximum production values were indeed detected. Authors should comment on that.
Section 2.2: Syngas simulation. Why was this composition chosen as a representative of syngas? Syngas is defined as a mixture of CO and H2.
Results
Figures 1 B and C should be changed since Figure 1 C is mentioned in the text first.
“It is important to note that varying the gas to liquid ratio impacts on growth and product formation more than varying the bottle orientation during agitation.” In order to state that you should have tested 4:1 horizontally. You have compared 9:1 with 4:1, when you tested the gas to liquid ratio. Your control was 4:1 vertical. Therefore, to investigate if the bottle orientation had a comparable effect you should compare with the control. I believe that when you used 9:1 you achieved the maximum gas transfer and, in this case, using the bottle vertically or horizontally was indifferent. You don’t have to do experiments, but I believe that you cannot use this sentence.
Lines 276-277: name of microbial species should be underlined since all the text is already in italics. Check for this mistake in other parts of the document.
Lines 285-286: name of microbial species should be in italics. Check for this mistake in other parts of the document.
Figures 1,2 and 3: why cell growth is presented as OD600 since in materials and methods section a conversion factor to CDW is specified? I believe that cell concentration in g/L is important to evaluate results.
Figure 2: Which is the composition of syngas? It should be in the legend.
The term “optimized” for the medium that was composed by mineral elements of the “Basal” medium and the trace metals and vitamins of the “Enriched” medium is not appropriate because for engineers this term is related to maximization or minimization of a function. Therefore, it could lead the reader to think that an optimization procedure was performed to obtain the composition of this medium.
“Our study presented a multifaceted optimization campaign” Again I believe that the term “optimization” is not appropriated and should be removed from the whole manuscript.
Figures 1,2,3,4,5,6: fermentation time for maximum production should be mentioned in the legends.
Author Response
We assembled the answers to the Reviewer's comments in the enclosed file.

Reviewer 2 Report
Overview: The manuscript describes the production of 2,3-BDO by Clostridium ljungdahlii and Clostridium autoethanogenum from various combinations of gas substrates. The study is interesting because the authors showed the gas mixtures that can enhance either the conversion of gases to 2,3-BDO or ethanol. The manuscript is well written; however, the authors should also address the following concerns:
- The authors should italicize throughout the manuscript the names of the microorganisms used in the study and those that were mentioned in the ‘Result and Discussion’ section.
- The authors argued that the higher Zn and Fe in the ‘Optimized + Fe’ medium enhanced 2,3-BDO titer and productivity. Why is it that the ‘Enriched’ medium failed to enhance 2,3-BDO titer and productivity as with the ‘Optimized’ medium because both contain the same trace element solution with similar mineral solution except that the ‘Enriched’ medium contains ~1.3-, ~9- and ~2-fold higher KCl, PO4 and Ca, respectively, with ~10-fold less NaCl? Or does it mean that a higher concentration of KCl, PO4 or Ca negatively impacts 2,3-BDO production? It is certain that Ca enhances 2,3-BDO based on information in the literature. This needs more clarification in the manuscript.
Author Response
We assembled the answers to the Reviewer's comment in the enclosed file.

Round 2
Reviewer 1 Report
The term "optimized" is still in the title of the manuscript and should be replaced
All other corrections have been performed and now the manuscript can be accepted for publication.
Author Response
Dear Reviewer,
we'd like to let you know we applied the change required in the manuscript.
